# Sensory experience steers representational drift in mouse visual cortex

Joel Bauer [1,2,6,10] ✉, Uwe Lewin [1,3,10], Elizabeth Herbert [4], Julijana Gjorgjieva [4], Carl E. Schoonover [5,7], Andrew J. P. Fink [5,8], Tobias Rose [1,9], Tobias Bonhoeffer [1] & Mark Hübener [1] ✉

Representational drift—the gradual continuous change of neuronal representations—has been observed across many brain areas. It is unclear whether drift is caused by synaptic plasticity elicited by sensory experience, or by the intrinsic volatility of synapses. Here, using chronic two-photon calcium imaging in primary visual cortex of female mice, we find that the preferred stimulus orientation of individual neurons slowly drifts over the course of weeks. By using cylinder lens goggles to limit visual experience to a narrow range of orientations, we show that the direction of drift, but not its magnitude, is biased by the statistics of visual input. A network model suggests that drift of preferred orientation largely results from synaptic volatility, which under normal visual conditions is counteracted by experience-driven Hebbian mechanisms, stabilizing preferred orientation. Under deprivation conditions these Hebbian mechanisms enable adaptation. Thus, Hebbian synaptic plasticity steers drift to match the statistics of the environment.

A growing body of evidence indicates that stimulus-evoked neuronal responses change gradually over days to weeks, even in the absence of any experimental manipulation[1–7]. This phenomenon, now referred to as representational drift[8,9], varies widely in magnitude across sensory and associative cortices, as well as stimulus types[2–7,10].

Little is known about the causes of representational drift. Is it the consequence of activity-independent, and therefore experience-independent, synaptic volatility[11–13], or does it reflect the effect of experience-dependent synaptic changes[14,15]? If drift is caused by synaptic volatility alone, its dynamics should resemble a random walk in stimulus representation space[8,16]. Conversely, if drift is a function of experience, its dynamics may be directed to reflect that experience. Several independent observations in different brain regions have indicated that drift can be sensitive to an animal's experience between recordings. For example, in the piriform cortex, the daily experience of an odorant halves the drift rate to that particular stimulus[5]. Conversely, recent studies of factors contributing to drift in spatial tuning of hippocampal place cells have reported a destabilizing effect of experience[17,18].

Historically, neuronal responses in primary sensory areas have been considered relatively stable in adult animals. In the primary visual cortex (V1), the stability and perturbation resistance of large-scale functional maps for retinotopy, orientation, and ocular dominance[19–22] suggest that these basic tuning features are stably encoded. More recent chronic recordings of single-neuron activity in the mouse have confirmed this high level of stability and perturbation-resistance for certain aspects of V1 responsiveness, including preferred orientation (PO), ocular dominance, and spatial frequency preference[23–26]. Other measures, however, such as tuning curves[3,27], especially for responses to complex stimuli and natural movies[10,27], have been found to drift to varying degrees.

[1]Max Planck Institute for Biological Intelligence, Martinsried, Germany. [2]International Max Planck Research School for Molecular Life Sciences, Martinsried, Germany. [3]Graduate School of Systemic Neurosciences, Ludwig-Maximilians-Universität München, Planegg, Germany. [4]School of Life Sciences, Technical University of Munich, Freising, Germany. [5]Mortimer B. Zuckerman Mind Brain Behavior Institute, Department of Neuroscience, Columbia University, New York, NY, USA. [6]Present address: Sainsbury Wellcome Centre for Neural Circuits and Behaviour, University College London, London, UK. [7]Present address: Allen Institute for Neural Dynamics, Seattle, WA, USA. [8]Present address: Department of Neurobiology, Northwestern University, Evanston, IL, USA. [9]Present address: Institute for Experimental Epileptology and Cognition Research, University of Bonn, Medical Center, Bonn, Germany. [10]These authors contributed equally: Joel Bauer, Uwe Lewin. ✉e-mail: joel.bauer@ucl.ac.uk; mark.huebener@bi.mpg.de

The PO of a mouse V1 neuron–the contour orientation that elicits the strongest response–has previously been reported to be relatively stable[23–26]. This is puzzling in light of continuous synaptic changes in V1 (observed as dendritic spine turnover[28–30]), and reports that orientation tuning curves measured on separate days become progressively more dissimilar with time[3,10,27]. A columnar organization for orientation preference, which exists in the visual cortex of many mammals, could place anatomical constraints on PO drift by ensuring that the set of synapses forming onto and retracting from a cortical neuron prefer the same orientation[7]. However, rodents lack clear orientation columns[31], raising the question of how PO could be stable. This makes the PO of neurons in mouse V1 an ideal model for studying the effect of experience on drift.

Here, we use PO in mouse V1 to determine whether the dynamics of representational drift depend on the animal's ongoing experience of its visual environment. We hypothesize that PO stability emerges from animals' frequent experience of oriented contours during the normal visual experience and that the consequent coactivation of similarly tuned neurons maintains the existing connectivity in the network against a background of continuous synaptic volatility[5,12]. Using chronic two-photon calcium imaging in adult mice, we first show that PO undergoes modest drift in V1 neurons. We then restrict the range of orientations that the mouse experiences[20,32,33] for several weeks by applying cylinder lens goggles that limit visual input to contours of one orientation for several weeks[34–36]. We find that this manipulation biases drift towards the remaining experienced orientation, but does not affect drift magnitude. Finally, we build a network model of PO drift and show that in this model, experience steers drift to match the input statistics via Hebbian plasticity. This mechanism stabilizes PO under normal visual conditions by counteracting the destabilizing effect of synaptic volatility.

## Results
### PO of V1 neurons drifts over time
To characterize drift in the orientation tuning of V1 neurons, we first quantified how the tuning of single cells changes over time. To this end, we presented gratings moving in 12 directions (six orientations) to awake, adult mice on 12–15 days over the course of a month while measuring neuronal activity using two-photon imaging of layer 2/3 cells expressing GCaMP6s[24] (Fig. 1a; Supplementary Fig. 1a). All mice were exposed to the full stimulus set on at least two separate days before the start of the experiment to avoid confounding effects from stimulus-response attenuation[37]. We found that the similarity of neuronal responses between experimental sessions decreased as a function of time (Fig. 1b; Supplementary Fig. 1b), similar to previous reports of representational drift in the visual cortex[3,10,27].

We hypothesized that this decay in tuning curve similarity is at least partly caused by shifts in the PO of neurons. Indeed, we found neurons whose tuning curves shifted over days to weeks (Fig. 1c; Supplementary Fig. 1c). To quantify these changes, we calculated the PO for each repeatedly identified and responsive neuron (see Methods; Supplementary Fig. 2) on each day as the vector sum of responses to the different orientations. Using a trial resampling approach (bootstrapping, see Methods), we determined the 95% confidence interval of this PO measure for each day. If the confidence interval of a neuron was above 45°, it was considered untuned (i.e., was considered to have no PO) and excluded from further analysis for that day. We determined the PO change of a neuron across two days to be significant if the PO of each respective day was outside of the confidence interval of the other day. On the population level, the POs of neurons appeared more similar on two consecutive days compared to 20 days apart (Pearson's correlation: $r = 0.968$ and $0.80$ respectively; Fig. 1d). Quantifying this observation across sessions revealed that both the relative number of significant PO changes (Fig. 1e) and the magnitude of these PO changes (Fig. 1f, g) increased over time. Nevertheless, the

overall PO drift rate was low, with a median drift rate of -0.3°/day (calculated from all 19–20 day intervals), in line with findings that basic visual tuning features are rather stable and show only limited representational drift[10,25,26].

Previous studies have raised the question of whether representational drift may be due, at least in part, to changes in neuronal firing caused by differences in animal behavior or state between sessions rather than to actual changes in neural tuning[38,39]. In V1, behavioral variables such as running have been shown to modulate the activity of neurons[40,41]. However, running or behavioral state have negligible effects on PO, specifically[40,42–45]. Thus, PO drift should be largely robust to this confound. Indeed, while we observed slight variability in PO between running and non-running trials within a single session, it was too small to explain PO drift over several days (Fig. 1h; in line with ref. 45). Additionally, we quantified running and arousal modulation indices for each neuron and scaled these by the change in running or arousal across session, in order to obtain an estimate of the effective modulation by behavioral state. Excluding the 50% of PO changes that were most affected by behavioral state changes had little effect on the overall PO drift (Supplementary Fig. 1d–g). The PO drift we observed is, therefore, unlikely to be explained by behavioral changes across days.

### Visual experience steers PO drift
Representational drift of PO is moderate compared to drift in other brain areas[2,4,5]. A potential factor that could stabilize responses to a stimulus is how often it is experienced by the animal[5]. Therefore, we hypothesized that the PO of neurons in V1 is stabilized by the frequent occurrence of oriented contours during normal vision. To test this, we used cylinder lens goggles[34–36] to deprive mice of experiencing all but one orientation (Fig. 2a). In the past, changes in neural representations observed during baseline conditions have been referred to as drift, while changes during or after experimental manipulations (e.g., monocular deprivation) were referred to as experience-dependent plasticity or learning[8]. In this study, we will refer to all time-dependent PO changes as drift, including the ones occurring during orientation deprivation. This is because we consider representational drift to encompass such experience-dependent changes in representations and hypothesize similar synaptic mechanisms to underlie both phenomena.

We found that applying the goggles continuously for four weeks biased the distribution of POs towards the experienced orientation (Fig. 2b; Supplementary Fig. 3a–c; two-sample $T$ test, t(6)2.85, $p = 0.029$), consistent with single time point recordings in juvenile mice[35,36]. Using longitudinal imaging, we characterized the change of individual neurons' PO over the course of the experiment. In our analyses, we excluded neurons that gained or lost orientation tuning (252 and 171 neurons, respectively, from a total of 835), focusing on changes in PO. However, we found that the PO of neurons that gained orientation selectivity over the course of the experiment tended to be near the experienced orientation, though this effect was not significant (Supplementary Fig. 3d, e; one-sample $T$ test on change in cell number with relative PO between 0° and 45° t(6)2.1, $p = 0.080$).

Defining the PO relative to the experienced orientation as rPO, we calculated the overall PO convergence of the population towards the experienced orientation (convergence: Δ|rPO|). We found that, on average, the PO of neurons drifted towards the experienced orientation (Fig. 2c; Supplementary Fig. 4a). Notably, the size of this PO convergence was dependent on the duration of orientation deprivation (Fig. 2c).

This convergence towards the experienced orientation could be explained by two possible factors: an experience-dependent change in the magnitude of PO drift (Fig. 2d), and/or a bias in the direction of PO drift towards the experienced orientation (Fig. 2e). Note, that drift magnitude does not take the direction of the PO change into account,

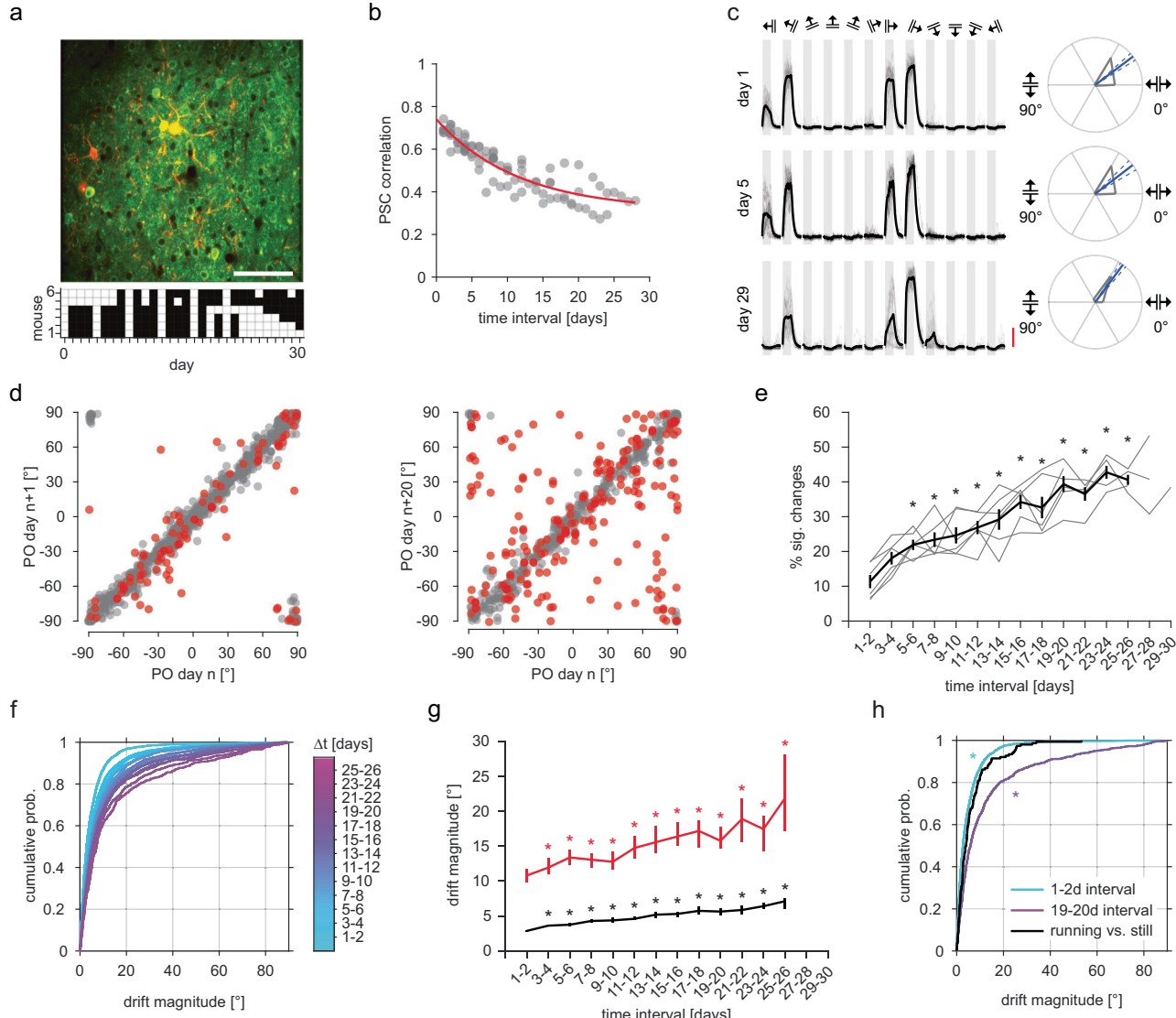

**Fig. 1 | Preferred orientation of V1 neurons drifts over time. a** Example field of view from one mouse. Top: Average GCaMP6s (green) and mRuby2 (red) fluorescence, scale bar 100 μm. Bottom: imaging timeline of six mice, white squares are days with imaging sessions. **b** Correlation of the pairwise signal correlation (PSC) matrices compared across all imaging sessions of one mouse. Red: exponential decay fit ($y = 0.31 + 0.43e^{-0.09*x}$). **c** Left: responses of an example neuron on 3 days. Single-trial responses, gray, average black. Gray bars: stimulus window of 5 s, grating directions indicated above. Scale bar: 100 ΔF/F. Right: polar plots of the responses on the left, with gray lines indicating mean response. Preferred orientation (PO) in blue with 95% confidence intervals (CI) as blue dashed lines. **d** Left: POs 1 day apart. Two-sided circular-circular Pearson's correlation $r = 0.968$ ($p < 1 \times 10^{-16}$), $n = 781$ PO changes from 169 neurons from six mice. Right: POs 20 days apart. $r = 0.800$ ($p < 1 \times 10^{-16}$), $n = 360$ PO changes from 170 neurons from six mice. Red: significant changes, gray: non-significant changes. **e** Percentage of concurrently tuned cells that significantly changed their PO vs. interval length. Six individual mice as gray lines, with black line as mean with error bars as S.E.M. One-

way ANOVA F(61)14.98, $p = 5.62 \times 10^{-14}$. Asterisks indicate Dunnett's post hoc test (1–2 days vs. all), $p < 0.05$. **f** Cumulative probability distributions of the absolute size of PO changes (|ΔPO|; drift magnitude) for different intervals. **g** Median drift magnitude for all PO changes in black and only significant PO changes in red. Error bars are bootstrapped 95% CIs. Two-sided Kruskal–Wallis test on all PO changes ($\chi^2(12)$ 864, $p = 2.9 \times 10^{-177}$, $n = 468–4194$ PO changes), and on only significant PO changes ($\chi^2(12)$ 122, $p = 2.4 \times 10^{-20}$, $n = 188$ to 347 PO changes). Asterisks indicate Bonferroni corrected two-sided Mann–Whitney U tests between 1 and 2 days vs. all other intervals $p < 0.05$. **h** Cumulative probability distribution of drift magnitude. Cyan: 1–2 day intervals, $n = 2147$ PO changes from five mice; magenta: 19–20 day intervals, $n = 657$ PO changes from five mice; black: changes between still trials and running trials within sessions, $n = 152$ PO changes from five mice. Two-sided Mann–Whitney U test for within-session changes between running and still trials compared to changes across short time intervals ($U = 9.78 \times 10^6$, $p = 1.04 \times 10^{-5}$), or long time intervals ($U = 1.11 \times 10^6$, $p = 2.13 \times 10^{-8}$). Source data are provided as a Source Data file.

while convergence does. We found that the initial PO of neurons and their drift magnitude were not correlated (Fig. 2f), and the overall PO drift magnitude of the population was largely unaffected by the altered visual experience (Supplementary Fig. 4b). Furthermore, shuffling drift magnitude across the population while leaving cell-wise drift directions intact had little effect on the overall convergence (Fig. 2g). On the other hand, most neurons' PO drifted towards the experienced orientation rather than away from it (Fig. 2h) and shuffling the drift

directions, while keeping cell-wise drift magnitudes intact, abolished the population convergence entirely (Fig. 2i; Supplementary Fig. 4c). Together, this demonstrates that depriving neurons of their PO does not lead to larger drift magnitudes, but rather steers their drift direction towards the experienced orientation.

If the absence of experience of most orientations during deprivation is the cause of PO drift towards the remaining experienced orientation, restoration of normal vision should recover the initial PO

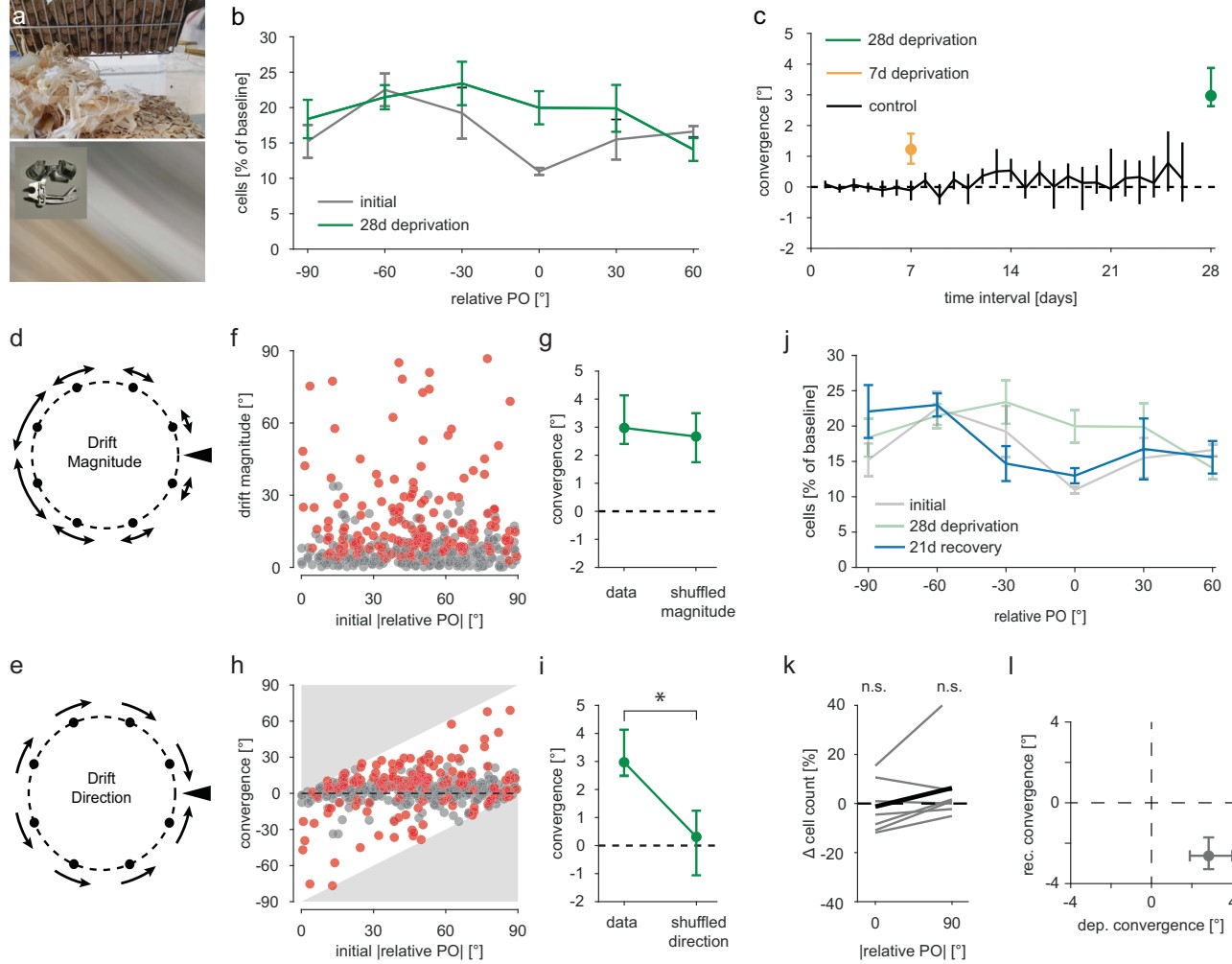

**Fig. 2 | Orientation deprivation leads to PO convergence. a** Photograph of a visual scene in a mouse cage taken without (top) and with (bottom) cylinder lens goggles (inset). Note that only one orientation is present in the bottom image. **b** Distributions of POs relative to the experienced orientation (at 0°; relative PO: rPO) before (gray) and after (green) 28d of orientation deprivation. Cell numbers are shown as a percentage of the initial total cell count (835 neurons from seven mice). Mean across mice with S.E.M error bars. **c** Median convergence (Δ|rPO|) for different time intervals under normal visual conditions (black, six mice) and orientation-deprived visual conditions (orange, eight mice deprived for seven days; green, seven mice deprived for 28 days). Positive Δ|rPO| indicates changes towards the experienced orientation. Error bars are bootstrapped 95% CIs. **d** PO convergence could result from neurons whose PO is similar to the experienced orientation (black triangle) drifting less than those with more dissimilar POs. **e** Alternatively, neurons may drift towards the experienced orientation, while drift magnitude is unaffected. **f** Initial PO difference from experienced orientation is uncorrelated with drift magnitude (|ΔrPO|) after 28-day deprivation. Two-sided Spearman's correlation r = −0.034 (p = 0.950, n = 414 neurons from seven mice).

**g** Median convergence (Δ|rPO|) before and after shuffling drift magnitudes. Error bars are bootstrapped 95% CIs. Two-sided Wilcoxon signed rank z = 0.807, p = 0.419, (n = 414 neurons from seven mice). **h** Initial PO difference from experienced orientation plotted against convergence (Δ|rPO|). n = 414 neurons from seven mice. Gray areas indicate impossible values. **i** Median convergence (Δ|rPO|) before and after shuffling drift directions. Error bars are bootstrapped 95% CIs. Two-sided Wilcoxon signed rank z = 4.19, p = 4.94 × 10⁻⁵ (n = 414 neurons from seven mice). **j** Distributions of POs relative to the experienced orientation (0°): initial (gray), after 28d orientation deprivation (green), and after 21d of recovery (blue). Cell numbers are shown as a percentage of the initial total cell count. N = 7 mice. Mean across mice with S.E.M error bars. **k** Relative change in cell numbers from initial to post recovery, for two PO bins (±0° to ±45° and ±45° to ±90° from the experienced orientation). Gray lines are seven individual mice, black line is mean. Two-sided one-sample T tests were used to compare within bin changes (t(6) −0.309, p = 0.768 and t(6)0.998, p = 0.357). **l** Median PO convergence during orientation deprivation vs. during recovery with 95% CI (n = 335 neurons from seven mice). Source data are provided as a Source Data file.

distribution. Further, intermittent exposure to all orientations should reduce the convergence effect. Indeed, we found that exposing the awake mice to drifting gratings during imaging for a few hours every seven days resulted in substantially lower convergence compared to uninterrupted orientation deprivation (Supplementary Fig. 4d). We also found that after ending deprivation, the initial distribution of POs was largely recovered, as the population drifted in the opposite direction than during the deprivation phase (Fig. 2j–l; Supplementary Fig. 4e).

Taken together, these data show that while the magnitude of PO drift of neurons in adult mouse V1 is unaffected by experience, the

direction of PO drift is determined by the distribution of experienced orientations. Thus, PO drift is experience-dependent.

## Hebbian plasticity and synaptic volatility can explain experience-dependent PO drift

To understand the nature of synaptic plasticity, which could lead to experience-dependent PO drift, we built a computational model of a neural network subject to synaptic plasticity (Fig. 3a). Previous studies have shown that synaptic changes can occur both in the presence and the absence of neuronal activity or NMDA-receptor mediated plasticity[13,46–48]. These activity-independent synaptic fluctuations are

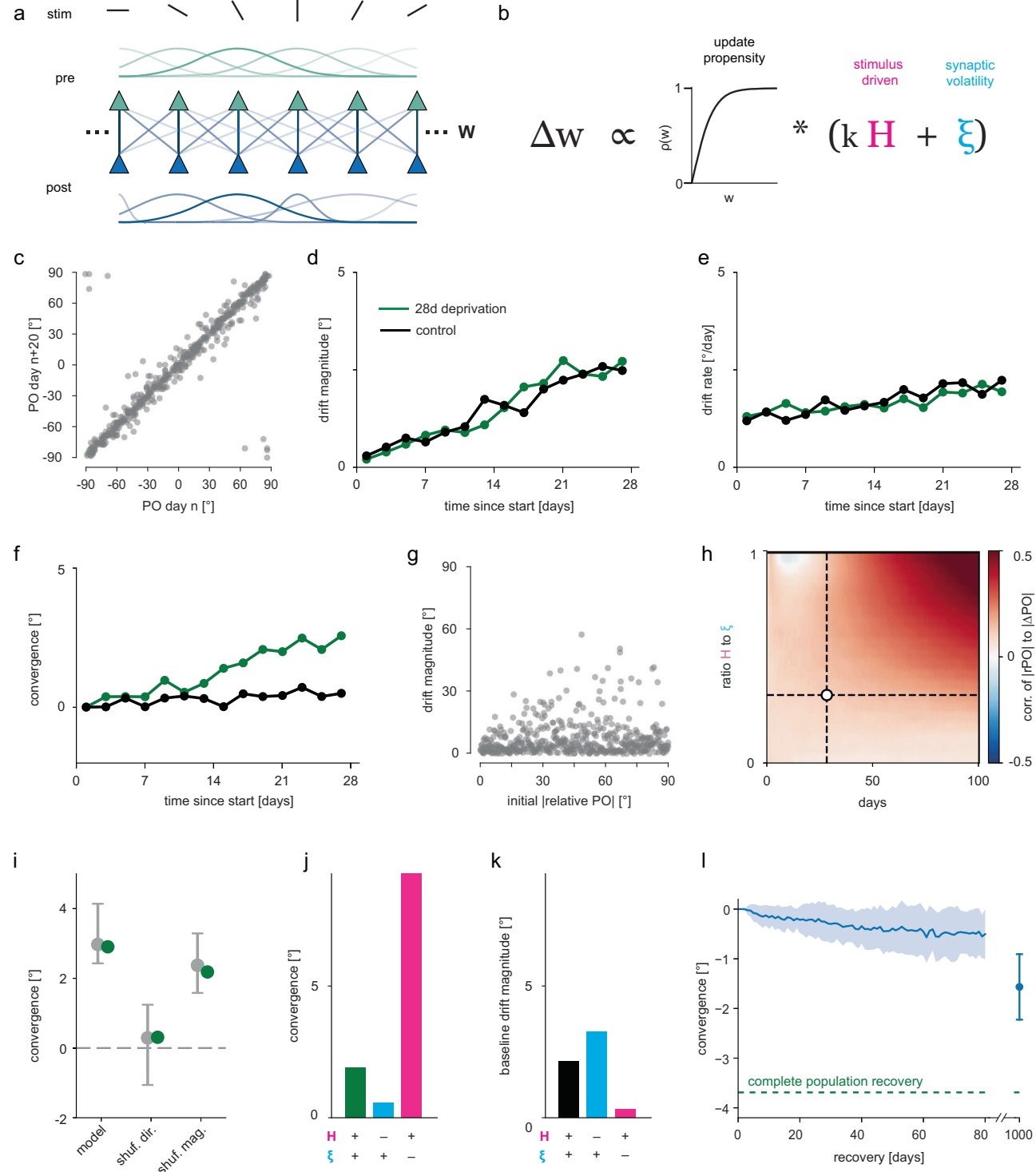

referred to as intrinsic synaptic volatility[12,49], here, synaptic volatility. We, therefore, reasoned that the observed PO drift might be driven by a plasticity rule combining experience-dependent Hebbian ($H$) changes and experience-independent synaptic volatility ($\xi$). We modeled the change in feedforward synaptic weights from a cortical layer of presynaptic neurons to a cortical layer of postsynaptic neurons as the sum of $H$ and $\xi$ scaled by a synaptic weight-dependent propensity function ($\rho(w)$): $\Delta w = \rho(w)(kH + \xi)$ (Fig. 3b). This propensity function was inspired by experimental results showing that the magnitudes of changes in spine size—commonly considered a proxy for synaptic strength—is proportional to the initial size of the spines[47,50–53]. We

initialized the feedforward weights from orientation-tuned presynaptic neurons as circularly-symmetric Gaussian distributions with varying widths, resulting in orientation-tuned postsynaptic neurons with varying tuning widths (Fig. 3a; Supplementary Fig. 5a–d). We modeled the experience-driven component $H$ as Hebbian changes—i.e., the product of presynaptic and postsynaptic activities—in response to a series of orientation stimuli, and $\xi$ as a baseline random synaptic change. The weights were subject to homeostatic normalization on a slow timescale[54–56] (four orders of magnitude slower than Hebbian changes; see Methods) to preserve the total sum of input weights per neuron.

**Fig. 3 | Network model indicates PO drift is a trade-off between Hebbian plasticity and synaptic volatility. a** Model schematic: two-layer network with $n$ orientation-tuned presynaptic neurons (*pre*) connected topographically with weights ($w$) to $n$ orientation-selective postsynaptic neurons (*post*). $n = 500$. **b** Synaptic plasticity rule: weights ($w$) are updated in proportion to the sum of stimulus-driven changes ($H$; scaled by $k$) and synaptic volatility ($\xi$), both scaled by a propensity factor proportional to initial weight ($\rho(w)$). **c** POs of postsynaptic neurons in the model 20 days apart, under baseline stimulus conditions. **d** Median drift magnitude ($|\Delta PO|$) increases over time, and is comparable under baseline (black) and orientation-deprived (green) conditions. **e** Same as **d** but for mean drift rate ($PO_{day}-PO_{day-1}$). **f** Median convergence ($\Delta|rPO|$) over time for baseline (black) and orientation-deprived (green) conditions. **g** Initial distance from experienced orientation ($|rPO|$) shows little correlation with drift magnitude after 28-day deprivation. Two-sided Spearman's correlation $r = 0.118$ ($p = 0.008$, $n = 500$). **h** Spearman's correlation between drift magnitude and initial distance from experienced orientation (as shown in **g**) in networks with different ratios of Hebbian plasticity to synaptic volatility ($H/\xi$; see **b**). Synaptic volatility is constant, while the Hebbian component is scaled by $k$. Correlation increases with longer deprivation time. Large Hebbian plasticity component also leads to increased correlation. Dashed lines indicate $H/\xi$ ratio in the other panels and 28-day deprivation length. **i** Shuffling drift direction but not magnitude abolishes the median convergence effect of the model during orientation deprivation. Green: model data and shuffles ($n = 500$). Gray: experimental data from Fig. 2g, i, median convergence and shuffles. Error bars are bootstrapped 95% CIs; $n = 414$ neurons from seven mice. **j** Effect of omitting either the Hebbian or the synaptic volatility contribution from the plasticity rule on PO convergence, during 28 days of orientation-deprived conditions. **k** Effect of omitting either the Hebbian or the synaptic volatility contribution from the plasticity rule on PO drift magnitude, during 28 days of baseline stimulation conditions. **l** The model displays limited recovery after input statistics return to baseline conditions. During recovery, median convergence is negative and slowly increases in magnitude over time (blue), but is incomplete even after 1000 days (~3 years). Mean (solid line/data point) and standard deviation (shaded region/error bars) over 50 model iterations. Source data are provided as a Source Data file.

To model the experience of mice under baseline conditions, reflecting normal vision, we stimulated the network with orientations sampled uniformly from −90° to 90°. A 'day' in our model consisted of one orientation stimulus per second for 12 hours. The model produced PO drift similar to our experimental data, with the POs of neurons drifting progressively over time (Fig. 3c, d; Supplementary Fig. 5e). To mimic the effect of orientation deprivation using cylinder lens goggles, we stimulated the network with a single orientation, instead of all orientations. Here, POs slowly drifted towards the experienced orientation (Fig. 3f). Neurons that are tuned far from the experienced orientation have weak connections with presynaptic neurons close to the experienced orientation (Supplementary Fig 5a), and these are not strengthened much due to the propensity function, raising the question of how these neurons are able to drift towards the experienced orientation. We find that these neurons shift because the synapses that are already strong and close to the neurons' own PO are strengthened and that among these potentiated synapses, there is an asymmetry favoring synapses that are closer to the experienced orientation (Supplementary Fig 5f–h). However, day to day drift rate—not measurable from experimental data directly—remained constant (Fig. 3e). As in the experimental data described above (Fig. 2f), drift magnitude was largely independent of the initial PO within 28 days of deprivation (Fig. 3g; Supplementary Fig. 5i, j). However, the correlation between drift magnitude and initial relative PO gradually increased with time (Fig. 3h). This suggests that the lack of correlation between drift magnitude and initial PO, within the timeframe tested experimentally (Fig. 2f), was caused by synaptic volatility masking the effect of Hebbian plasticity, with the associated correlation only becoming evident after much longer time intervals. The rate at which this correlation increased was also strongly dependent on the relative strength of the Hebbian component (Fig. 3h). Shuffling the magnitude or direction of PO changes produced the same effect as in our data (Fig. 3i), demonstrating that, both in experimental data and model, convergence of the population was due to a bias in drift direction. In the model, Hebbian changes drove directional PO changes and thus convergence towards the experienced orientation (Fig. 3j; Supplementary Fig. 6b), while synaptic volatility introduced non-directional drift (Fig. 3k; Supplementary Fig. 6a). Importantly, baseline drift was highest in the absence of Hebbian changes (Fig. 3k; Supplementary Fig. 6a). Hence, Hebbian plasticity compensates for synaptic volatility under baseline conditions, and steers drift when input statistics change.

Our model also aligned with the experimental effect of interruptions during the deprivation period. Brief exposure to the full range of orientation stimuli every seven days slowed down PO convergence by transiently removing the directional bias, while leaving drift magnitude unaffected (Supplementary Fig. 6c). However, the model exhibited much slower recovery on a population level than observed experimentally. After deprivation had ended, the population displayed a negative convergence (divergence) that grew with time (Fig. 3l), but this never reached the magnitude observed in the experimental data (Fig. 2k). Likewise, the population distribution only partially recovered from the effect of deprivation, even after almost 3 years (Supplementary Fig. 6d).

With this, our model yields two specific predictions about PO drift beyond what we have found experimentally. First, the slower recovery of the original PO distribution in our model suggests that the plasticity rule employed is insufficient to fully explain the observed recovery in V1. Additional mechanisms may be present in the mouse visual cortex that accelerate recovery, such as backbone spines (synapses with little or no plasticity)[8]. The second prediction is that removing the Hebbian contribution experimentally would result in progressively larger PO drift magnitudes, both during baseline and deprivation conditions (Fig. 3k; Supplementary Fig. 6a, b). Thus, our model strongly suggests that under conditions of normal visual experience, stimulus-driven activity acts via Hebbian plasticity mechanisms to limit the destabilizing effect of synaptic volatility.

## Discussion
### PO of V1 neurons undergoes representational drift
Previous studies have shown that representational drift occurs in the visual cortex, mainly by quantifying the decay in response correlations over time[3,10,27]. Some observations of drift have been explained by time-varying changes in behavior[38,39]. We establish drift in the PO of individual neurons, a canonical visual response property, and demonstrate that these changes accumulate over time irrespective of locomotion and arousal. Representational drift is thought to originate from an accumulation of synaptic weight changes and synaptic turnover, while further response variability is caused by behavioral variability, internal state fluctuations, or measurement noise[9]. Time-varying changes in behavior and signal-to-noise ratio can result in time-dependent tuning curve changes that can be mistaken for representational drift when measurements and state changes occur on similar timescales[39]. PO is well-suited to distinguish representational drift from such variability for two reasons. First, in contrast to tuning to other visual features, PO is state-independent, i.e. it is unaffected by running or behavioral state[40,42–45]. Second, a decrease in signal-to-noise ratio reduces the accuracy of PO estimation but does not cause PO to trend towards a specific value. This is in contrast to direction selectivity, for instance, where a decrease in signal-to-noise would cause a tendency towards lower estimates of selectivity. The confounding effect of decreases in signal-to-noise ratio can, therefore, be mitigated by applying a constant confidence threshold on PO changes.

## What controls PO drift?

Isolating and quantifying drift in a fundamental visual response property, PO, allowed us to explore the mechanisms that underlie drift. The cause of drift in neuronal tunings, such as the PO drift we observe, may be the accumulation of synaptic strength changes and turnover over time. Dendritic spine volume changes and turnover have been used in the past to estimate the extent of these changes and clearly demonstrate that some degree of change in connectivity does occur in mouse V1. However, the majority of spines persist for many weeks, and there is no consensus on the degree of baseline synaptic changes or its effect on neuronal response properties to date[28–30,57]. These synaptic changes, while often associated with activity-dependent synaptic plasticity[46,58–60], also arise from activity-independent synaptic volatility[11,12,47–49,61]. Accumulation of the apparently undirected activity-independent component should, in principle, lead to progressive changes in neuronal tuning. However, tuning curve similarity remains relatively high, i.e., the signal correlation asymptotes far above zero even after long time intervals[3,10,23–26], raising the question of what might limit the effect of synaptic volatility. Two potential mechanisms are anatomical constraints and compensatory plasticity[16,62,63].

Stability in neuronal tuning could arise from the anatomical constraints imposed by the spatial organization of axons and dendrites. The tuning properties of neurons depend on the availability of appropriate axons in the vicinity of their dendrites[64], which is largely determined during development, with little change observed during adulthood[65–67]. The position of a neuron relative to a cortical map is, therefore, expected to correlate with its tuning stability. In support of this, a recent study in mouse somatosensory cortex found that layer 2/3 neurons tuned to the topographically aligned whisker are more stable than neurons tuned to a surrounding whisker[7]. We speculate that in mammals that have orientation columns in V1, neurons at the pinwheel center, where more diverse inputs are available, should drift more than those within an iso-orientation column[68]. However, as there is little spatial organization of PO in rodent visual cortex[31], PO drift of V1 neurons in mice should not be constrained by the availability of inputs.

Hebbian plasticity rules suggest that the repeated experience of sensory stimuli can stabilize the connectivity between, and the tuning of, co-tuned neurons through their coincident activity, ultimately leading to slower drift[8,16]. This has been proposed for olfactory cortex, where responses to odor stimuli are more stable for odorants that are experienced more frequently[5]. We, therefore, hypothesized that orientation tuning in V1 could be stabilized by the frequent experience of all orientations. While we found that orientation deprivation had no observable effect on the magnitude of PO drift within the time frame tested, the direction of PO drift was biased towards the experienced orientation, resulting in an overall convergence of PO towards the remaining experienced orientation. This shows that visual experience steers PO drift rather than limiting its magnitude.

## Hebbian plasticity and synaptic volatility control PO drift

Using a computational network model, we found that a combination of synaptic volatility and Hebbian plasticity driven by visual experience recapitulated the experimental dynamics of PO drift both during normal vision and orientation deprivation. POs changed as a function of time, and PO convergence increased as a function of deprivation duration. There was also next-to-no correlation between the initial PO of modeled neurons and the drift magnitude during deprivation, within a timeframe corresponding to that of our experiments. Our experimental findings, both during normal experience and deprivation, are therefore compatible with the hypothesis that the dynamics of PO drift in the mouse visual cortex are driven by a combination of two synaptic processes, synaptic volatility and experience-dependent Hebbian plasticity. However, this was only possible because of the weight-dependent propensity function, which favored plasticity changes at stronger synapses.

Beyond demonstrating plausibility, our theoretical model makes several clear predictions. Completely removing the experience-dependent Hebbian component increases drift under baseline conditions but prevents convergence during orientation deprivation. Conversely, removing synaptic volatility reduces drift and allows faster convergence. The model fits our data best when the contribution of synaptic volatility exceeds Hebbian learning. These predictions could be tested in future experiments using our paradigm while manipulating the relative ratio of the two synaptic plasticity components, e.g., by transiently blocking NMDA receptor-dependent plasticity.

In the absence of the experience-dependent Hebbian component, our model predicts that PO drift would increase. A recent study found that PO changes were higher during a baseline ~8-day interval of normal vision compared to a subsequent ~8-day interval of dark exposure[69]. The authors concluded that experience has a destabilizing effect on tuning stability. This contrasts with both our experimental and modeling findings. It may, however, be partially explained by the sequential design of the dark exposure study. Over the first few successive visual stimulation sessions, neural responses tend to decrease and become sparser[37]. This could lead to larger tuning differences between the control and dark exposure imaging sessions in the study by Jeon et al.[69]. However, uniform dark exposure and partial visual deprivation very differently affect input statistics, and the resulting correlation structure of activity generates different outcomes[70]. We, therefore, cannot exclude that there are additional mechanisms, not contained within our network model, which are engaged during dark exposure that cause lower drift magnitudes.

We suggest that experience-dependent synaptic changes exert a net stabilizing force on visual representations in the presence of synaptic volatility. This is similar to the stabilizing effect of the repeated experience of the same olfactory stimulus in piriform cortex[5]. In contrast, recent experiments in the CA1 region of mouse hippocampus have shown that repeated experience of the same environment leads to drifting spatial tuning[17,18], whereas the passage of time alone largely affects response magnitude[18]. A higher rate of Hebbian plasticity at hippocampal synapses may explain these results, as ongoing experience could rapidly induce random exploration of a solution space, as it occurs for networks trained using stochastic gradient decent[71,72]. Alternatively, the difference in the rate and experience-dependence of drift between areas may be due to differences in the dimensionality, or degrees of freedom, of the representations[72]. Whatever the reason, these recent data, together with our own results, add to the growing body of work showing that the phenomenon of representational drift, as well as its mechanistic underpinnings, need to be interpreted in the context of the specific computation, network architecture, processing hierarchy, molecular cell type, and the ethological relevance of a circuit[72,73].

## Methods

### Animals

All experimental procedures related to animal handling were carried out in compliance with institutional guidelines of the Max Planck Society and using protocols approved by the Regierung von Oberbayern (Beratende Ethikkommission nach §15 Tierschutzgesetz). Female wild-type C57BL/6NRj mice (Janvier) were used for all experiments in this study. Mice were housed at $55 \pm 5$ % humidity and $22 \pm 1.5 \,°C$ under a 12-hour inversed light-dark cycle with food and water available ad libitum. Mice were housed in large cages (GR900, Tecniplast) with access to a running wheel and other enrichment material such as a tunnel and a house. Animals were usually group-housed, except for short time periods during recovery from surgery.

### Cranial window implant and intrinsic optical signal imaging targeted virus injections

In order to repeatedly measure visual tuning properties of neurons in mouse V1, neurons were virally transduced with the genetically

encoded calcium indicator GCaMP6s, and a cranial window implanted over the visual cortex. Mice between P35 and P45 were anesthetized using a mixture of Fentanyl (0.05 mg/kg), Midazolam (5 mg/kg), and Medetomidine (0.5 mg/kg), injected intraperitoneally. Anaesthesia was maintained by injecting 25% of the original dose after the first two hours and then every hour. All surgical equipment was heat sterilized using a bead sterilizer and rinsed with ethanol. After checking for anaesthesia depth using the toe pinch reflex, the mice were placed on a thermostatically controlled heating blanket (set to 37 °C), and the head was fixed in a stereotaxic frame. Carprofen (0.5 mg/kg) was administered subcutaneously as an analgesic. Throughout the procedure, the eyes were kept moist and protected from debris using eye cream (Isopto-Max). The scalp was disinfected using iodine and ethanol, and lidocaine was applied as a local anaesthetic. The scalp was then removed along with the periosteum, and the hair along the rim of the wound was fixed using Histoacryl. The skull was roughened using a scalpel, and a custom aluminum head bar was fixed onto the skull using superglue (Pattex Ultra Gel). Dental cement (Paladur) was used to further secure the head-bar to the skull, cover the edges of the scalp wound, and cover the exposed skull (except for where the craniotomy would be placed). A rim of cement was built up around the front of the craniotomy in order to form a well. Diluted ultrasonic gel (diluted with cortex buffer in a ratio of 3:1; cortex buffer: in mM: 125 NaCl, 5 KCl, 10 glucose, 10 HEPES, 2 $CaCl_2$, and 2 $MgSO_4$) was placed in this well, and a 10 mm cover glass place over it, making sure to prevent bubbles. This increases the optical transparency of the skull for subsequent intrinsic optical signal (IOS) imaging.

The mice were then moved to an IOS imaging setup. An image of the blood vessel pattern over the visual cortex was taken with 530 nm illumination light (using a tandem 135 mm $f$/2.0 and 50 mm $f$/1.2 objective system, pco.edge 4.2 LT sCMOS camera or a Thorlabs B-Scope equipped with a ×4 objective and a 1500 MonoChrome camera). The skull was then illuminated with 735 nm light, and the focal plane lowered to ~400 nm below the surface of the brain. A 700–740 nm bandpass filter was placed in front of the camera sensor. The contralateral eye was covered with a cone made of tape and drifting gratings were presented 7–14 times on a screen 16 cm in front of the mouse (8 directions at 2 cycles/s and 0.04 cycles/° changing direction every 0.6 s, presented for 7 seconds each time, with a horizontal retinotopic position spanning 0 to −30° azimuth). The resulting intrinsic signal changes were analyzed using custom-written MATLAB code. This allowed the localization of V1, to aid targeting of virus injections.

The mice were then returned to the stereotaxic frame, and a 4 mm craniotomy was performed using a high-speed micro drill with the intended virus injection and two-photon imaging locations at the center. The skull was kept moist using cortex buffer during drilling. A 4 mm bone patch was removed, and the exposed cortex (still covered by dura) was washed repeatedly with cortex buffer until any bleeding stopped. The brain was covered with Gelfoam pieces soaked in cortex buffer to keep it moist and protected from debris.

Before the surgery, borosilicate glass injection pipettes were pulled (using a PC-97 pipette puller). Volumetric markings were drawn onto the borosilicate capillaries before pulling (marks were drawn every 1 mm using a black permanent marker, corresponding to 45 nl). The pipette tips were broken off so that the tip diameter was between 25 and 35 μm and beveled to a sharp point using a modified computer hard disk (Canfield 2006). The pipettes were front-loaded with the virus by placing a drop of virus on parafilm, lowering the pipette tip into the virus drop, and applying negative pressure. The pipettes were either loaded with a mixture of AAV2/1.1CamKII0.4.Cre.SV40 (titer 1-5 E8), AAV2/1.Syn.Flex.mRuby2.GSG.P2A.GCaMP6s.WPRE.SV40 (titer 1.28 E13) and AAV2/1.Syn.GCaMP6s.WPRE.SV40 (titer 6 E12), or AAV2/1.Syn.mRuby2.GSG.P2A.GCaMP6s.WPRE.SV40 (titer 1.2 E13). The loaded pipette tips were kept on ice until use.

The pipette was placed into a patch pipette holder and inserted 300–400 μm into the cortex using a micromanipulator. After waiting 4–5 min, virus was injected at ~50 nl/min with a total volume of 150–250 nl. This was done using 30–40 psi with 20–40 msec pressure pulses at 0.8 Hz controlled by a pulse generator (Master-8) and a pressure micro-injection system (Toohey Company). The pipette was left in the cortex for another 4-5 min before retracting it. The process was repeated for 2–5 injection sites, with a spacing of 50–100 μm. Throughout the injection procedure, the cortex was kept moist using cortex buffer. After all injections were completed, the cortex was covered with a 4 mm cranial window (glass coverslip), secured in place using super glue (Pattex Ultra Gel) around the rim of the craniotomy. After allowing the glue to dry, dental cement was used to cover any remaining bone and further secure the glass cover slip.

The mice were then administered 0.5–1 ml Sterofundin subcutaneously, and anesthesia was antagonized using Naloxone (1.2 mg/kg), Flumazenil (0.5 mg/kg), and Atipamezole (2.5 mg/kg) also administered subcutaneously. The animals were kept in a warm environment and observed for several hours before returning to their home cage. Wet food was placed into the cages, and 0.5 mg/kg of Carprofen was administered for the first 3 days after surgery. Fluorophores were allowed to express for 2–3 weeks before checking expression. If excessive bone regrowth prevented imaging, a second short surgery was performed to replace the cranial window and remove any bone patches.

After surgeries, the mice were placed in a 14/10 h light dark-cycle reversed room, so that the animals could be imaged during their dark cycle. Mice were co-housed with conspecifics (littermates whenever possible) in a large 1500 $cm^2$ cage containing nesting material, dark retreats and a running wheel. These conditions were upheld during orientation deprivation.

## IOS imaging and Cylinder lens goggle mounting
2–4 weeks after the virus injection, IOS imaging was repeated (as described above) but without the use of ultrasound gel and using a drifting and inverting checkerboard bar (Fourier stimulus) to map out the retinotopic gradient across V1[74]. This stimulus allowed for better outlining of the V1 border. In these experiments, mice were anaesthetized only lightly, with Fentanyl (0.035 mg/kg), Midazolam (3.5 mg/kg) and Medetomidine (0.35 mg/kg), injected intraperitonially. Anesthesia was maintained by injecting 25% of the original dose every hour. For animals that would later undergo orientation deprivation, after IOS imaging custom aluminum goggle frames (without cylinder lenses) were fitted to the animal's head and adjusted using tongs. The space between the frames and the eyes was such that the animals could still clean their eyes with their forepaws. The goggle frames were then attached to the head bars with small screws.

## Chronic in vivo two-photon calcium imaging in awake mice
Prior to awake head fixation, mice were handled for five days to accustom them to the experimenter, walking on a Styrofoam ball with a fixed axis and brief head restraint. For imaging, mice were head-fixed on an air-suspended Styrofoam ball, allowing them to run, under a two-photon imaging system (Thorlabs Bergamo II). The two-photon system was equipped with a pulsed femtosecond Ti:Sapphire laser (Spectra Physics MaiTai DeepSee laser; tuned to 940 nm for calcium imaging or 1050 nm for structural imaging), resonant and galvo scanning mirrors, a ×16 NA 0.8 immersion objective (Nikon) and a piezoelectric stepper for multiplane imaging. The photon collection pathway had a 720/25 nm short-pass filter followed by a dichroic beam-splitter (FF560) that allowed simultaneous detection of green and red light using two GaAsP photomultiplier tubes (PMTs; Hamamatsu) with either a 500–550 nm or a 572–642 nm bandpass filter. The system was controlled by ScanImage 4.2[75]. Ultrasound gel (diluted 3:1 with water and centrifuged to remove air bubbles) was applied over the cranial

window implant, and the ×16 objective was immersed in the gel. Strips of tape were used to shield the imaging window from external light.

During all imaging sessions, videos of both eyes were recorded (Imaging Source infra-red camera, 30 Hz), and running was tracked via an infra-red sensor attached to the air-suspended Styrofoam ball. The pupil diameters were extracted using DeepLabCut[76] and averaged across both eyes. The running/arousal modulation for each PO change (i.e., for each neuron across each pair of sessions) was defined as the correlation coefficient between the stimulus-response amplitude of the neuron and the running speed/pupil diameter of the mouse, multiplied by the average change in running speed/pupil diameter across sessions, i.e.:

$$running\ modulation = corr(running\ speed, stimulus\ response\ amplitude) \\ * \overline{\Delta running\ speed} \quad (1)$$

$$arousal\ modulation = corr(pupil\ diameter, stimulus\ response\ amplitude) \\ * \overline{\Delta pupil\ diameter} \quad (2)$$

Mice were presented with visual stimuli on a gamma-corrected LCD monitor (60 Hz, 2560 × 1440 pixels, 27 inches) 16 cm from the eyes. Light contamination during two-photon imaging was minimized by shuttering the LCD monitor[77]. The screen was either placed directly in front of the mouse if imaging binocular V1 or offset when imaging monocular V1, and the screen tilt was aligned to the angle of the mouse's head. The same screen position and screen angle were used for repeated sessions of each mouse. Visual stimuli were generated in MATLAB using the Psychophysics Toolbox[78,79]. The visual stimulus consisted of a 25° radius (+12° fading-edge) sine wave grating drifting in 12 directions (0.04 cycles/° and 3 cycles/s) on a gray background with a stimulus duration of 5 s and an interstimulus interval of 6 s (gray screen). The mice were presented with these stimuli on two separate days before data acquisition began as part of habituation. On data acquisition days, the stimuli were presented 32 times. During these days, appropriate fields of view were identified and imaged during stimulus presentation allowing for re-finding on subsequent days. We re-found this imaging field of view on subsequent days by searching for the matching blood vessel pattern under widefield illumination. After switching to two-photon imaging, we used custom-written MATLAB code to overlay the live field of view over a template (average fluorescence image) acquired during the first imaging session. This alignment was significantly aided by the tdTomato structural marker. We then adjusted the live position in X, Y, and Z until the two images were perfectly aligned. During the experiment, if there was any slow drift in depth this was manually corrected by the experimenter. Again, this process was made significantly easier by the tdTomato structural marker.

## Orientation deprivation

Mice that underwent orientation deprivation were habituated to the goggle frames for a minimum of four days before the baseline imaging sessions. The goggle frames were removed when presenting visual stimuli and recording neuronal calcium signals and were again fixed to the animal's head bar after the end of the imaging session. When imaging was followed by orientation deprivation, the cylinder lenses were pressed into the goggles (sometimes secured with super glue) before attaching them to the animal's head bar again. Mice were subsequently returned to their home cage (which had stripes of black tape oriented along the experienced orientation through the cylinder lenses). Alternatively, mice were housed in a cage with monitors around two sides of the cage that presented slowly drifting full field gratings moving in eight directions during the light cycle. The experienced orientation through the cylinder lenses was between −22° to −45°.

## Processing of in vivo two-photon calcium imaging data and calculation of POs

Fluorescence traces from in vivo data were extracted either using custom MATLAB software or Suite2P[80]. Both methods produced average fluorescence FOV images, single neuron ROI average fluorescence over time, with ROIs independently calculated or drawn for each day. To match ROIs from different sessions, we used a custom-written MATLAB program, which registered the templates from different sessions using affine transformation. If ROIs had more than 50% overlap in their ROI masks, they were defined as putatively matched. We then inspected the ROIs of each neuron manually. If any of the ROIs of a neuron were not in the same location relative to local landmarks (other neurons, blood vessels, axons, dendrites) or the neuron was no longer clearly visible in one of the imaging sessions, the neuron was excluded from the dataset. Only neurons that could be reidentified on every imaging session were included in further analysis. We verified this approach by employing an additional quantitative method. To this end, we calculated pixel-wise correlations of ROIs for a subset of our data. We plotted the distribution across all comparisons as-well-as the minimum correlation for each neuron across all time points (Supplementary Fig. 2a). Excluding neurons with a minimum ROI correlation below 0.1, 0.25 or 0.5 did not substantially change the overall drift magnitude (Supplementary Fig. 2b–e).

The calcium activity of a neuron was defined to be $\Delta F/F$ and was calculated as:

$$\Delta F/F = \frac{(F(t) - 0.7*(F_{neuropil}(t) - \widetilde{F}_{neuropil})) - \widetilde{F}_{baseline}}{\widetilde{F}_{baseline}} \quad (3)$$

Where $F(t)$ is the green fluorescence trace of a neuron, $F_{neuropil}(t)$ is the neuropil green fluorescence trace, $\widetilde{F}_{neuropil}$ is the median of the neuropil fluorescence trace and $\widetilde{F}_{baseline}$ is the median of all 1 s pre-stimulus periods of the ROI fluorescence[24].

The stimulus tuning curve of a neuron was defined as the mean $\Delta F/F$ during the stimulus presentation across all trials for a given grating direction (resulting in a vector length 12). To calculate the pairwise signal correlation (PSC) matrix, the Pearson's correlation between the stimulus tuning curves of all pairs of neurons were calculated (resulting in a square matrix size neurons x neurons). The pairwise stimulus correlation matrix of each day was vectorized and correlated with the vectorized PSC matrix of every other day (resulting in a square matrix size days x days).

For further analysis, the PO was calculated as follows. First, neurons were tested for visual responsiveness. For each stimulus, a rank sum test between the stimulus period mean $\Delta F/F$ and the pre-stimulus period mean $\Delta F/F$ was applied (with Bonferroni correction significance threshold of $p < 0.05$). If a neuron's $\Delta F/F$ was significantly above the $\Delta F/F$ during the pre-stimulus period for any stimulus, it was considered visually responsive. Second, the PO of each neuron was calculated using the vector sum across all responses:

$$X = \frac{1}{n_{rep}*n_{stim}} \sum_{rep=1}^{n_{rep}} \sum_{stim=1}^{n_{stim}} R_{(stim,rep)} \cos(2\,\theta_{(stim,rep)}) \quad (4)$$

$$Y = \frac{1}{n_{rep}*n_{stim}} \sum_{rep=1}^{n_{rep}} \sum_{stim=1}^{n_{stim}} R_{(stim,rep)} \sin(2\,\theta_{(stim,rep)}) \quad (5)$$

$$\theta_{PO} = \frac{1}{2} \arctan\left(\frac{Y}{X}\right) \quad (6)$$

Where $R_{(stim,rep)}$ is the stimulus period mean $\Delta F/F$ per stimulus per trial, $\theta_{(stim,rep)}$ are the corresponding visual stimulus angles, $n_{stim}$ is the number of different stimulus angles and $n_{rep}$ is the number of stimulus

repetitions (usually 32). Bootstrap resampling across all stimulus repetitions was applied in order to obtain a distribution of $\theta_{PO}$. PO is defined as the $\overline{\theta_{PO}}$ across all bootstrap samples. The 95% confidence interval (upper $PO_{CI}$ and lower $PO_{CI}$) was also calculated from the distribution of $\theta_{PO}$. If the difference between the upper $PO_{CI}$ and lower $PO_{CI}$ was greater than 90°, then the neuron was considered not orientation-tuned. A neuron was considered concurrently tuned across a time interval, if it was both visually responsive and tuned on both days. For these neurons the ΔPO was calculated. In order to determine if the PO of a cell was significantly different on two separate days, the PO of the second day had to be outside the PO 95% confidence interval of the first day, and the PO of the first day had to be outside the PO 95% confidence interval of the second day.

The experienced orientation during orientation deprivation was calculated by taking an image of the mouse, wearing the goggle frames, from the center of the stimulus presentation screen. The angle of the goggles in the image was measured using ImageJ. This angle was adjusted based on the angle of the image presentation screen to give the experienced orientation relative to the mouse. To calculate PO convergence, the change in a neuron's PO relative to the experienced orientation (i.e., if it moved towards or away from the experienced orientation, *experienced*), was calculated as follows:

$$\text{for mice without goggles}: \Delta|\text{rPO}| = |\text{PO}_{t1} - \overline{\text{experienced}}| \\ - |\text{PO}_{t2} - \overline{\text{experienced}}| \quad (7)$$

$$\text{for orientation deprivation data}: \Delta|\text{rPO}| \\ = |\text{PO}_{pre} - \text{experienced}_{mouse}| - |\text{PO}_{post} - \text{experienced}_{mouse}| \quad (8)$$

Where $\overline{\text{experienced}}$ is the mean experienced orientation through the cylinder lens goggles of all orientation deprived mice, and $\text{experienced}_{mouse}$ is the experienced orientation for the mouse from which the neuron came.

## Statistics
When normality could not be ruled out using the Kolmogorov−Smirnov Goodness-of-Fit test, two-tailed two-sample unequal-variance, Pearson's correlation, one-way ANOVA, paired or unpaired $T$ tests were used. When normality was ruled out or could not be assumed, Spearman's correlation, the Wilcoxon signed-rank test, the Mann−Whitney $U$ test, or the Kruskal−Wallis test were used. In the case of periodic variables, a von Mises distribution was assumed, and circular statistics were used (circ_stats toolbox[81]). If ANOVA or Kruskal−Wallis tests were significant, we used a Dunnett's posthoc test when data were parametric, or Mann−Whitney $U$ tests with Bonferroni corrected alpha thresholds when data were not parametric. Animal number and cell or PO change numbers are indicated by capital and lower case "$n$", respectively, in the figure legends. Asterisks indicate $p$ values under significance threshold $\alpha = 0.05$ (*). Alpha values were adjusted using the Bonferroni method for multiple comparison correction where necessary.

## Network model
We considered a feedforward network in which $N = 500$ presynaptic excitatory neurons with firing rates **u** drive $N$ postsynaptic excitatory neurons with firing rates **v**, via a matrix of synaptic connections **W** and a linear response function such that $\mathbf{v} = \mathbf{W}^T\mathbf{u}$. Presynaptic neurons were assumed to be tuned to a given orientation from 0° to 180°, and their firing rates were modeled with Gaussian tuning curves centered at the orientation to which they were tuned. In the model, we assumed that the drift in orientation tuning seen in our data can be mainly explained by changes to the feedforward connectivity

between excitatory neurons in two cortical layers of the primary visual cortex.

**Learning rule.** We found that the representational drift observed in our data, both under baseline and deprivation conditions, can be captured by three main assumptions:
1. Synaptic changes consist of a combination of both activity-dependent synaptic plasticity in response to external stimuli, represented by a Hebbian component $H$, and activity-independent fluctuations in synaptic strength (synaptic volatility), represented by a random component $\xi$.
2. Synaptic changes are scaled by a weight-dependent propensity function, $\rho(w)$.
3. Synaptic weights are normalized by homeostatic mechanisms acting on a slower timescale than the timescale of stimulus-driven Hebbian plasticity.

We combined these assumptions into a learning rule which describes the change in the synaptic strength $w_{ij}$ connecting pre-synaptic neuron $u_j$ to post-synaptic neuron $v_i$ as a weighted sum of Hebbian $H$ and random $\xi$ changes, scaled by a weight-dependent propensity function $\rho(w)$:

$$\Delta w_{ij} = \epsilon \cdot \rho(w_{ij}) \cdot \left[ kH_{ij} + \xi \right] \quad (9)$$

The learning rate $\epsilon$ ($1 \times 10^{-4}$) scales the overall magnitude of the weight update, and the scaling factor $k$ determines the relative impact of the Hebbian and random contribution to the total weight changes in the model. We modeled the propensity function to be weight-dependent $\rho(w) = \tanh(10w)$ based on experimental observations[47,50–53]. The Hebbian contribution $H_{ij}$ for each weight $w_{ij}$ was given by the product of presynaptic and postsynaptic activities in response to each stimulus presentation:

$$H_{ij} = u_j v_i \quad (10)$$

The random component $\xi$ represents activity-independent synaptic strength fluctuations, and was sampled independently for each weight $w_{ij}$ from a standard normal distribution at each weight update:

$$\xi \sim N(0, 1) \quad (11)$$

The weights were normalized on a timescale slower than Hebbian activity-dependent plasticity, motivated by experimental findings of slower homeostatic mechanisms such as synaptic scaling[54–56]. Following the presentation of $N_\theta$ stimuli, we implemented a divisive normalization which preserves the total sum of incoming weights $w_{ij}$ onto each postsynaptic neuron $i$:

$$w_{ij} \leftarrow \frac{w_{ij}}{\sum_j w_{ij}} \quad (12)$$

Given that homeostatic plasticity is typically reported to occur on a timescale of hours to days[55], we assumed that this normalization occurs once per day. Assuming each pyramidal neuron in the mouse visual cortex experiences approximately one orientation stimulus per second, we took $N_\theta$ to be (60 seconds) × (60 minutes) × (12 waking hours) = 43,200 stimuli per day. We then scaled the overall learning rate $\epsilon$ to match the mean drift magnitude to that observed over the experimental time frame.

**Weight initialization.** The population of $N$ orientation-selective pre-synaptic neurons had Gaussian-shaped tuning curves uniformly distributed across the space of oriented stimuli from −90° to 90°. The input weights to the $N$ postsynaptic neurons were initialized as

Gaussians across the space of orientations, such that each postsynaptic neuron inherited an initial tuning curve from its presynaptic partners (Supplementary Fig. 5a). The drift rate of a model neuron was correlated with the width of its initial tuning curve (Supplementary Fig. 5d). In the experimental data, the drift magnitudes were approximately log-normally distributed across neurons. To match this variability seen in drift magnitude, we, therefore, initialized the postsynaptic tuning curves with widths sampled from a log-normal distribution (Supplementary Fig. 5a, b).

**PO and drift metrics.** The PO of neurons in the network model was calculated as follows: a series of orientation stimuli from −90° to 90° were presented to the network, and the PO of each neuron was defined as the orientation that elicits the highest firing rate in the postsynaptic neurons. Weights were frozen during this test protocol and were not subject to synaptic plasticity. Drift magnitude and convergence were defined as in the experimental analysis, and drift rate was defined as the absolute circular distance between POs on consecutive days.

**Visual experience conditions.** To simulate the experience of a mouse during baseline visual conditions, we repeatedly stimulated the network with stimuli drawn uniformly between −90° to 90°. To mimic orientation deprivation conditions, we stimulated the network instead with a constant experienced orientation $\hat{\theta}$ (value selected randomly). For the deprivation condition, as in the experiment, the network was either stimulated constantly with the same orientation for 28 days, or interrupted every 7 days by a test protocol in which a repeated series of stimuli across the entire range of −90° to 90° were randomly presented. Weights were plastic and continuously updated by the exact same learning rule during both baseline and deprivation conditions, unless otherwise stated (Fig. 3h, j, k; Supplementary Fig. 6a, b). For all experiments, a short warm-up period (2–3 days) of baseline conditions was run to allow the weights to settle to a constant distribution before measurement began.

**Ratio of Hebbian plasticity to synaptic volatility contribution.** Network parameters (amplitude and offset of presynaptic firing rates, overall weight scaling) were set such that when $k = 1$, the mean values of $H_{ij}$ and $\xi$ across neurons were equivalent under baseline conditions. To modulate the relative strength of the Hebbian component with respect to synaptic volatility (Fig. 3h), we therefore scaled $k$ from 0 to 1. To determine the individual contributions of the Hebbian and random component to the drift (Fig. 3j, k), we removed each component respectively.

**Recovery from deprivation.** To test the recovery of orientation tuning in the network after deprivation (Fig. 3l), we simulated $N_d$ days of deprivation conditions, followed by $N_b$ days of baseline conditions, maintaining the same parameters throughout. The total directional PO changes undergone during the deprivation period were then compared to those during the following baseline period.

**Reporting summary**
Further information on research design is available in the Nature Portfolio Reporting Summary linked to this article.

## Data availability
The processed data are available at https://gin.g-node.org/Joel-Bauer/orientation_tuning_drift. Source data are provided with this paper.

## Code availability
The code is available at https://github.com/Joel-Bauer/orientation_tuning_drift and https://github.com/betsyherbert/bauer-lewin-drift-2024/tree/main.

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

## Acknowledgements

We thank Sandra Reinert for discussions and comments on the manuscript, Pieter Goltstein for help with the two-photon microscope and programming, Volker Staiger, Claudia Huber, Frank Voss, Dominik Lindner, and Max Sperling for technical assistance. We also thank Sasha Hay, Ezgi Kaya and Meike Hack for help with image annotation. This project was funded by the Max Planck Society, the German Research Foundation (Grant SFB870 A07, project number 1188 03580, to T.R., M.H., and T.B.), and the European Research Council under the European Union's Horizon 2020 research and innovation program (grant agreement no. 804824 to J.G.).

## Author contributions

All authors contributed to the conceptualization of the study. J.B. and U.L. conducted the experiments and analyzed the data. E.H. implemented the model with input from J.G., J.B., and U.L. All authors discussed the results and wrote the manuscript.

## Funding

## Competing interests

The authors declare no competing interests.
