## [Peer Review File · Nature Communications]

REVIEWER COMMENTS

Reviewer #1 (Remarks to the Author):

In this manuscript, Bauer et al. investigate the interaction between Hebbian plasticity and ongoing synaptic turnover in response to manipulations of sensory input. The experiments are very nicely done, the analysis is clear, and the manuscript is careful and well written. I have a few concerns noted below, but overall I believe this paper will be a welcome addition to the field.

Major:

1) My principal comment is that I am not sure if I understand how one of their key findings—that the preferred orientation exhibits similar convergence regardless of distance from the experienced orientation—can be explained only by Hebbian plasticity and random volatility.

For a neuron with PO close to the experienced orientation, this makes sense. The synapses tuned to the experienced orientation would be strengthened via Hebbian plasticity, which would lead to a shift in PO for both subthreshold activity and ultimately the neuron's spiking output.

However, for a neuron with initial PO far from the experienced orientation, the strengthened synapses would all be tuned to orientations far from the PO. As a result, you would not expect the tuning of the threshold activity or spiking to subtly shift towards the PO.

The authors' model primarily considers network-level integration, and not subcellular integration, so I suspect this detail is getting lost somehow, but it is important for understanding the findings.

Minor:

1) The data shown in Fig 2b show reduced representation of the experienced orientation prior to deprivation. Was this done on purpose or was it just happenstance? It seems possible that the increased representation of this orientation after deprivation could be due to regression to the mean during ongoing turnover (though Fig 2j indicates that this is unlikely). In any case, it would be helpful to show examples from individual mice in a supplementary figure to give a sense of the changes observed in individual animals.

2) The longitudinal tracking of neurons across days was not thoroughly described, and accurate tracking is critical for their experiments. Ideally, some sort of threshold for ROI similarity other than manual curation could be used to exclude cells that are not well-tracked.

3) A suggestion: for longitudinal imaging, transgenic indicator expression is preferred over virus, since the expression is much more stable. Ramping AAV expression can cause changes in the AP>Ca²⁺ relationship, which can in turn cause progressive shifts in tuning (e.g., by weak responses becoming more apparent), and thereby cause overestimates of orientation drift. Using doxycycline

to control expression during development appears to eliminate overexpression issues with transgenics, if that was a concern. To be clear, I don't expect the authors to re-do the experiments by any means, this is just a consideration for follow-up studies.

4) The manuscript cites Grutzendler et al. (line 57) to support their statement about ongoing synaptic turnover in visual cortex. However, this paper found that there is remarkably little synaptic turnover in adult visual cortex (96% of spines stable for >1 month). The authors might discuss this more, as there is not a clear consensus over the degree of spine turnover in adult visual cortex.

5) Typo: extra comma after "conditions" on line 145.

Reviewer #2 (Remarks to the Author):

There has been a great deal of interest recently in representational drift of stimulus encoding in a variety of brain areas. The origin of this phenomenon, and how it relates to classic experience dependent plasticity, has however remained unclear. Using chronic two-photon imaging and restriction of visual experience to a single orientation, Bauer et al. show that in the mouse visual cortex the magnitude of representational drift is independent of experience, and therefore likely driven by intrinsic synaptic volatility. In contrast, the direction of drift is shaped by experience in a Hebbian fashion.

These are exciting and important results. My main concerns are twofold:

1) The distribution of preferred orientations (PO) at baseline is very odd. In Fig. 2B, it is normalised to the experienced orientation (0°). Unless I overlooked the relevant information, the range of experienced orientations (resulting from the orientation of the cylinder lenses) is never stated, but assuming this covered a wide range, I would have expected the 'initial' line in Fig. 2B to be more or less flat. Alternatively, if the experienced orientations were near vertical or horizontal, one might even have expected a peak at 0° since the cardinal orientations tend to be slightly overrepresented. But here there is a distinct trough at 0° and peaks at 60° either side. Even the experience induced shift in PO does not result in the 28d distribution peaking at 0° , nor does it have its minimum at -90° . One might argue that the 28d distribution looks more like a baseline than the 'initial'. What was the initial total cell count?

2) It is obviously critical for this kind of study that individual neurons are correctly identified across time points. The statement in the Methods (lines 721-22), "The ROIs were matched manually across days using custom written Matlab code. Only neurons that could be reidentified on every imaging session were included in further analysis" therefore is not sufficient. What were the criteria for matching across days and for reidentification?

In addition, there are a few minor points.

- Line 116: "Correlation of pairwise signal correlation (PSC)" – what is a correlation of a correlation? This should be reworded.

- Line 154: “we excluded neurons that gained or lost orientation tuning”: a statement of what percentage of neurons gained or lost orientation tuning would be helpful.
- Line 155: “we found that the PO of neurons that gained orientation selectivity over the course of the experiment tended to be near the experienced orientation” – what about cells that lost orientation selectivity?
- Lines 231-32: “The weights were subject to homeostatic normalization on a slow timescale” – what does that mean? How slow relative to the Hebbian changes?
- Line 254: “negative convergence” – what does that mean?

Point-by-point response

We would like to thank the reviewers for their time and efforts. The comments and suggestions were very helpful, and we have revised the manuscript accordingly. In short, we have incorporated several additional data visualizations/analyses to the supplementary figures and added clarifications to the methods and discussion.

Reviewers' comments are indicated in **bold**; quotes from the main manuscript in *italics*.

Reviewer 1:

In this manuscript, Bauer et al. investigate the interaction between Hebbian plasticity and ongoing synaptic turnover in response to manipulations of sensory input. The experiments are very nicely done, the analysis is clear, and the manuscript is careful and well written. I have a few concerns noted below, but overall I believe this paper will be a welcome addition to the field.

Major:

1) My principal comment is that I am not sure if I understand how one of their key findings—that the preferred orientation exhibits similar convergence regardless of distance from the experienced orientation—can be explained only by Hebbian plasticity and random volatility.

We thank the reviewer for this opportunity to clarify our point: We do not claim that the preferred orientation (PO) exhibits similar convergence regardless of the distance from the experienced orientation. We show that the drift magnitude is not correlated with the initial PO in the data and the model (within 28 days, Figs. 2f, 3g). We also show that this lack of correlation between the initial PO and drift magnitude is congruent with Hebbian plasticity and random synaptic volatility. We assume the reviewer is referring to this and will answer accordingly below, but we would first like to clarify the distinction which we have also emphasized in the main text in line 164-165:

“Note, that drift magnitude does not take the direction of PO change into account, while convergence does.” When all neurons change randomly (in direction and magnitude), magnitude is uniform across PO. However, this does not mean that if magnitude is uniform, changes must be random in direction. The fact that we see no correlation between relative PO and magnitude (Fig. 2f), therefore, only excludes the hypothesis that drift magnitude is experience-dependent (Fig. 2d), but leaves open the possibility that drift direction is experience-dependent (Fig. 2e). This is confirmed by the shuffling test we performed; wherein shuffling drift magnitude did not affect PO convergence while shuffling drift direction abolished it (Fig. 2g,i).

We do not claim that PO exhibits similar convergence regardless of distance from the experienced orientation, as we did not perform that analysis. This is because neurons tuned to orientations close to the experienced orientation cannot converge as much as those further away. For example, a neuron with a relative PO 5° from the experienced orientation can only converge by 5° , while a neuron at 85° relative PO can only get 5° further away (see shaded area in Fig. 2h). Therefore, the correlation between PO and convergence is difficult to interpret and we did not quantify it.

Why drift direction, but not magnitude, is experience-dependent is not immediately obvious, but rather counterintuitive. Therefore, we used a modeling approach to understand how different synaptic rules

could lead to such an effect. Our model suggests that this is because random synaptic volatility is uniform and masks the non-uniform changes driven by Hebbian plasticity. We now state this more clearly in line 248-251 *“This suggests that the lack of correlation between drift magnitude and initial PO, within the timeframe tested experimentally (Fig 2f), was caused by synaptic volatility masking the effect of Hebbian plasticity, with the associated correlation only becoming evident after much longer time intervals.”*

The effect of directed plasticity is evident in the data because shuffling drift direction abolishes all convergence (Fig. 2i). Our model shows that combining synaptic volatility and Hebbian plasticity can explain all of the effects we see in our data, except for the fast recovery (Fig. 3). But it is important that Hebbian plasticity and synaptic volatility are not, by themselves, sufficient. A crucial factor is the weight-dependent synaptic propensity function, which scales plasticity by the synapse's initial strength (Fig. 3b), and is based on experimental findings (Yasumatsu et al., 2008; Loewenstein et al., 2011; Berry and Nedivi, 2017; Humble et al., 2019; Statman et al., 2014). We will elaborate on this point below.

For a neuron with PO close to the experienced orientation, this makes sense. The synapses tuned to the experienced orientation would be strengthened via Hebbian plasticity, which would lead to a shift in PO for both subthreshold activity and ultimately the neuron's spiking output.

However, for a neuron with initial PO far from the experienced orientation, the strengthened synapses would all be tuned to orientations far from the PO. As a result, you would not expect the tuning of the threshold activity or spiking to subtly shift towards the PO.

The authors' model primarily considers network-level integration, and not subcellular integration, so I suspect this detail is getting lost somehow, but it is important for understanding the findings.

The reviewer is correct in asserting that for postsynaptic neurons tuned close to the experienced orientation, their synapses with presynaptic neurons closer to the experienced orientation are strengthened, allowing the postsynaptic neurons to shift their PO towards the experienced orientation. As the reviewer points out, this is not surprising. However, in our model, for neurons with a PO further away from the experienced orientation, it is not their synapses with presynaptic neurons orthogonally tuned to their own PO that are strengthened, as the reviewer suggests. Instead, it is those that are closer to the neuron's own PO. This is because of the weight-dependent propensity function (Figure 3b) and the fact that co-tuned inputs are stronger (Cossell et al., 2015). We have illustrated the effect of the propensity function in three additional supplementary figure panels (Supplementary Figure 4f-h). These panels show the contribution of Hebbian plasticity to the weight changes across the synaptic population before (**f**) and after (**g**) applying the weight-dependent propensity function. Neurons orthogonally tuned to the experienced orientation (dashed boxes in **f** and **g**) have weak synapses with presynaptic neurons close to the experienced orientation, which are not strengthened much. Instead, it is the synapses that are already strong and close to the neuron's own PO that are strengthened. Importantly, what causes these postsynaptic neurons' PO to shift is that among the strong synapses that are potentiated, there is an asymmetry favoring synapses that are closer to the experienced orientation compared to those that are further away, illustrated in **h**.

Supplementary Figure 4 f-h

- f** Mean Hebbian component H of synaptic weight changes accumulated over the 28-day deprivation period, calculated as the outer product of presynaptic and postsynaptic neuronal activities during this time window. Left and bottom bars: mean presynaptic and postsynaptic activity during the deprivation period. Pre- and postsynaptic neurons are sorted by their initial PO. Orange arrows indicate experienced orientation. Gray dashed box indicates synapses plotted in **h**.
- g** Similar to **f** but showing mean Hebbian influence on synaptic weight changes accumulated over the 28-day deprivation period, given by the Hebbian component weighted by the Hebbian strength k and the propensity function: $\rho(w) * [kH]$. Gray dashed box indicates synapses plotted in **h**.
- h** Hebbian influence for synapses between presynaptic neurons and postsynaptic neurons initially tuned far from the experienced orientation (gray box in **f** and **g**). Angle indicates PO difference between presynaptic and postsynaptic neuron. Synapses with a positive difference are closer to the experienced orientation than those with negative values. The distribution is biased towards synapses with presynaptic neurons tuned closer to the experienced orientation (circular mean: 4.64°). This allows postsynaptic neurons with POs far from the experienced orientations to progressively drift towards the experienced orientation.

We have therefore expanded the corresponding explanation in the main text in lines 238-245: “Here, POs slowly drifted towards the experienced orientation (Fig. 3f). Neurons that are tuned far from to the experienced orientation have weak connections with presynaptic neurons close to the experienced orientation (Supplementary Fig 4a) and these are not strengthened much due to the propensity function, raising the question of how these neurons are able to drift towards the experienced orientation. We find that these neurons shift because the synapses that are already strong and close to the neurons’ own PO are strengthened and that among these potentiated synapses, there is an asymmetry favoring synapses which are closer to the experienced orientation (Supplementary Fig 4f-h).”

Minor:

1) The data shown in Fig 2b show reduced representation of the experienced orientation prior to deprivation. Was this done on purpose or was it just happenstance? It seems possible that the increased representation of this orientation after deprivation could be due to regression to the mean during ongoing turnover (though Fig 2j indicates that this is unlikely). In any case, it would be helpful to show examples from individual mice in a supplementary figure to give a sense of the changes observed in individual animals.

We chose to use approximately the same cylinder lens goggle angle across all mice to avoid unbalanced experimental conditions across groups. We did not deliberately target the on average underrepresented $\sim 45^\circ$ orientation when choosing the angle of the cylinder lenses, this was indeed happenstance. We think

choosing a single angle was justified because our previous paper (Kreile et al., 2011) and a more recent study by Dias et al. (2022) showed that population shifts in orientation distributions occur regardless of the orientation of the goggles. As requested by the reviewer, we have now added the preferred orientation distributions before and after orientation deprivation relative to the head angle (and the corresponding experienced orientations) for each mouse to Supplementary Fig. 2a.

Supplementary Fig. 2a

a Distribution of true POs (relative to head angle) before (left) and after (right) deprivation for each mouse (colored lines) and on average (black line). Experienced orientation through cylinder lens goggles for each mouse is indicated by arrow heads. 835 neurons from 7 mice.

We do not think that regression to the mean due to ongoing synaptic turnover could explain our results. Regression to the mean would be akin to our model with just synaptic volatility but no Hebbian plasticity (Fig. 3j-k; Supplementary Fig. 4a-b). Based on the rate of drift during baseline and our modeling results, the PO shift magnitude we see would take an extremely long time and would also not explain why interrupted deprivation leads to much less convergence (Supplementary Fig. 3d). Most importantly, a population convergence due to regression to the mean would be invariant to PO drift direction while we find convergence is abolished when PO drift directions are shuffled.

Please also see our response to the first main point of reviewer 2.

2) The longitudinal tracking of neurons across days was not thoroughly described, and accurate tracking is critical for their experiments. Ideally, some sort of threshold for ROI similarity other than manual curation could be used to exclude cells that are not well-tracked.

For a detailed response to this important point please see our answer to the second main point of reviewer 2. In short, we have elaborated our description of the relevant methods in lines 723-729 and 743-748 of the revised manuscript.

3) A suggestion: for longitudinal imaging, transgenic indicator expression is preferred over virus, since the expression is much more stable. Ramping AAV expression can cause changes in the AP>Ca2+ relationship, which can in turn cause progressive shifts in tuning (e.g., by weak responses becoming more apparent), and thereby cause overestimates of orientation drift. Using doxycycline to control expression during development appears to eliminate overexpression issues with transgenics, if that was a concern. To be clear, I don't expect the authors to re-do the experiments by any means, this is just a consideration for follow-up studies.

We thank the reviewer for this suggestion. In our experiments, the first baseline imaging session included in the analysis was acquired over a month after virus injections, and we did not see any indications for ramping up of expression after this. But we agree with the reviewer that transgenic GCaMP expression alleviates this problem altogether.

4) The manuscript cites Grutzendler et al. (line 57) to support their statement about ongoing synaptic turnover in visual cortex. However, this paper found that there is remarkably little synaptic turnover in adult visual cortex (96% of spines stable for >1 month). The authors might discuss this more, as there is not a clear consensus over the degree of spine turnover in adult visual cortex.

Yes, the majority of dendritic spines are indeed persistent in mouse V1. We have added a short discussion and referenced a review where this is discussed more extensively, line 325-328: *“Dendritic spine volume changes and turnover have been used in the past to estimate the extent of these changes and clearly demonstrate that some degree of change in connectivity does occur in mouse V1. However, the majority of spines persist for many weeks, and there is no consensus on the degree of baseline synaptic changes or its effect on neuronal response properties to date (Grutzendler et al., 2002; Holtmaat et al., 2005; Majewska 2006; Runge et al., 2020).”*

5) Typo: extra comma after “conditions” on line 145.

We have removed the comma.

Reviewer 2:

There has been a great deal of interest recently in representational drift of stimulus encoding in a variety of brain areas. The origin of this phenomenon, and how it relates to classic experience dependent plasticity, has however remained unclear. Using chronic two-photon imaging and restriction of visual experience to a single orientation, Bauer et al. show that in the mouse visual cortex the magnitude of representational drift is independent of experience, and therefore likely driven by intrinsic synaptic volatility. In contrast, the direction of drift is shaped by experience in a Hebbian fashion.

These are exciting and important results. My main concerns are twofold:

1) The distribution of preferred orientations (PO) at baseline is very odd. In Fig. 2B, it is normalised to the experienced orientation (0°). Unless I overlooked the relevant information, the range of experienced orientations (resulting from the orientation of the cylinder lenses) is never stated, but assuming this covered a wide range, I would have expected the ‘initial’ line in Fig. 2B to be more or less flat. Alternatively, if the experienced orientations were near vertical or horizontal, one might even have expected a peak at 0° since the cardinal orientations tend to be slightly overrepresented. But here there is a distinct trough at 0° and peaks at 60° either side. Even the experience induced shift in PO does not result in the 28d distribution peaking at 0°, nor does it have its minimum at -90°. One might argue that the 28d distribution looks more like a baseline than the ‘initial’. What was the initial total cell count?

Thank you for raising this point, which was also highlighted by reviewer 1. We have now clarified the description of the initial and final PO distributions relative to the experienced orientations of the goggles. For context, the distribution of orientation tuning is known to be nonuniform across V1, with cardinal

orientations being overrepresented (Chapman and Bonhoeffer 1998; Girshick et al., 2011; Kreile et al., 2011; Li et al., 2022), and the local distribution of preferred stimulus directions and orientations depends on the retinotopic position in V1 (Fahey et al., 2019). In a previous publication we have shown that orientation deprivation leads to an increased representation of the experienced orientation regardless of the angle of the goggles (Kreile et al., 2011), and this has also been shown by others (Dias et al., 2022). We therefore chose to use only one angle of the goggles across all mice in order to avoid unbalanced experimental conditions across groups. Since our main claims are with respect to shifts in PO tuning of individual neurons rather than changes in population distributions alone, the initial distribution of the population is not crucial, only its change. But we agree that the underlying distributions were not made sufficiently clear in the original manuscript, and we have added an additional plot to Supplementary Fig. 2a showing the initial and final distributions of preferred orientations for each mouse in the 28-day deprivation group, aligned to the mouse head angle rather than the experienced orientation. The exact angle of the experienced orientation for each mouse is also indicated in these plots and we have added a clarification sentence to the methods line 738-739: “*The experienced orientation through the cylinder lenses was between -22° to -45°.*”. We have also applied the color code for mice to the rest of Supplementary Fig. 2 so that the population shift can be compared across these plots.

The number of orientation-tuned neurons at baseline was 583. 412 neurons maintaining PO tuning, 171 lost tuning and 252 gained tuning, giving a total of 835 neurons with orientation tuning either before or after deprivation. We have now indicated these cell numbers in the legends of Fig. 2 and Supplementary Fig. 2 and in the results text, line 154.

Supplementary Fig. 2a

a Distribution of true POs (relative to head angle) before (left) and after (right) deprivation for each mouse (colored lines) and on average (black line). Experienced orientation through cylinder lens goggles for each mouse is indicated by arrow heads. 835 neurons from 7 mice.

2) It is obviously critical for this kind of study that individual neurons are correctly identified across time points. The statement in the Methods (lines 721-22), “The ROIs were matched manually across days using custom written Matlab code. Only neurons that could be reidentified on every imaging session were included in further analysis” therefore is not sufficient. What were the criteria for matching across days and for reidentification?

We have elaborated the section in the methods to include a more detailed description of our neuron re-finding and ROI matching procedure.

- Alignment of field of view during imaging (added to line 723-729): *“We re-found these imaging field of views on subsequent days by searching for the matching blood vessel pattern under widefield illumination. After switching to two-photon imaging, we used custom written MATLAB code to overlay the live field of view over a template (average fluorescence image) acquired during the first imaging session. This alignment was significantly aided by the tdTomato structural marker. We then adjusted the live position in X, Y, and Z until the two images were perfectly aligned. During the experiment, if there was any slow drift in depth this was manually corrected by the experimenter. Again, this process was made significantly easier by the tdTomato structural marker.”*
- Matching neuron ROIs (added to line 743-748): *“To match ROIs from different sessions we used a custom written MATLAB program which registered the templates from different sessions using affine transformation. If ROIs had more than 50% overlap in their ROI masks they were defined as putatively matched. We then inspected the ROIs of each neuron manually. If any of the ROIs of a neuron were not in the same location relative to local landmarks (other neurons, blood vessels, axons, dendrites) or the neuron was no-longer clearly visible in one of the imaging sessions, the neuron was excluded from the dataset.”*

This was the same method used for several of our previous publications (Rose et al., 2016; Goltstein et al., 2021; Reinert et al., 2021). Currently, there is no gold standard for matching neuron ROIs over time, and some researchers use qualitative metrics based on human judgement (Rose et al., 2016; Marks & Goard 2022), while others use automated methods such as ROI percentage overlap of the mask or pixel correlation (Katlowitz et al., 2018; Sheintuch et al., 2017).

In order to test if employing an additional quantitative method would significantly change the magnitude of PO drift, we calculated pixel-wise correlation of ROIs for a subset of our data. We plotted the distribution across all comparisons as-well-as the minimum correlation for each neuron across all time points (see Rebuttal Fig. 1). Applying three different thresholds did not substantially change the overall drift magnitude.

Rebuttal Fig. 1

- a** Left: Distribution of image correlations between cropped areas around region of interest (ROI; as depicted in Supplementary Fig. 1a) for all pairs of matched ROIs. Right: Lowest ROI correlation for each neuron across all time intervals. Red dashed lines in a and b indicate cutoff thresholds of 0.1, 0.25 and 0.5.
- b** Fraction of significant PO changes (left) and median absolute PO changes (drift magnitude; right) for all PO changes in black and only significant PO changes in red (573 neurons from five mice).
- c** Same as **b** but excluding neurons with a minimum ROI correlation under 0.1 (8 of 573 neurons excluded).
- d** Same as **b** but excluding neurons with a minimum ROI correlation under 0.25 (25 of 573 neurons excluded).
- e** Same as **b** but excluding neurons with a minimum ROI correlation under 0.5 (161 of 573 neurons excluded).

In addition, there are a few minor points.

Line 116: "Correlation of pairwise signal correlation (PSC)" – what is a correlation of a correlation? This should be reworded.

This metric has been used in the past to quantify representational drift (Rose et al., 2016 and Montijn et al., 2016). First, the tuning curve correlation between all pairs of neurons is calculated for each imaging session, giving one matrix of correlation values per session. The correlation between these matrices is then calculated. In Fig. 1a this second order correlation is then plotted against the time interval between the sessions. This is explained in line 755-759. We have reworded line 115 (previous line 116) as concisely as possible to clarify what the first and second order correlations are "*Correlation of the pairwise signal correlation (PSC) matrices ...*".

Line 154: "we excluded neurons that gained or lost orientation tuning": a statement of what percentage of neurons gained or lost orientation tuning would be helpful.

We have added the following statement to the results section line 153-156 "*In our analyses we excluded neurons that gained or lost orientation tuning (252 and 171 neurons respectively from a total of 835 total), focusing on changes in PO.*" and specified these numbers in the Legends of Fig. 2 and Supplementary Fig. 2. Please see also our answer the next point.

Line 155: "we found that the PO of neurons that gained orientation selectivity over the course of the experiment tended to be near the experienced orientation" – what about cells that lost orientation selectivity?

We realize that our original Supplementary Fig. 2, which combined the cells gained and lost in one plot, was not sufficiently clear in this regard and have replaced it. We now show separate plots for the distribution of cells gaining and losing orientation selectivity in absolute numbers instead of relative changes in Supplementary Fig. 2d-e.

Lines 231-32: "The weights were subject to homeostatic normalization on a slow timescale" – what does that mean? How slow relative to the Hebbian changes?

The model is built such that homeostatic plasticity occurs once per day, whereas Hebbian plasticity happens once per stimulus presentation, which occurs once per second. See lines 834-838 in Methods. We have now clarified this in the results section line 231-233: “*The weights were subject to homeostatic normalization on a slow timescale (four orders of magnitude slower than Hebbian changes; see Methods) to preserve the total sum of input weights per neuron.*”

Line 254: “negative convergence” – what does that mean?

In the context of this paper convergence ($\Delta|rPO|$) is a metric ranging from -90° to 90° . While orientation deprivation induces positive convergence, recovery induces negative convergence. Negative convergence could also be called divergence but we thought this might be more confusing. We have added it in brackets nonetheless. Line 262-263: “*After deprivation had ended, the population displayed negative convergence (divergence) that grew with time (Fig. 3I),...*”

Rebuttal letter references

Berry, K. P. & Nedivi, E. Spine dynamics: Are they all the same? *Neuron* **96**, 43–55 (2017).

Chapman, B., Stryker, M.P. & Bonhoeffer, T. Development of orientation preference maps in ferret primary visual cortex. *J. Neurosci.* **16**, 6443-6453 (1996).

Cossell, L., Iacaruso, M., Muir, D., Houlton, R., Sader, E. N., Ko, H., Hofer, S. B., & Mrsic-Flogel, T. D. Functional organization of excitatory synaptic strength in primary visual cortex. *Nature* **518**, 399–403 (2015).

Dias, R.F., Rajan, R., Baeta, M., Marques, T. & Petreanu, L. Visual experience instructs the organization of cortical feedback inputs to primary visual cortex. *bioRxiv* (2022). doi: 10.1101/2022.10.12.511901

Fahey, P.G., Muhammad, T., Smith, C., Froudarakis, E., Cobos, E., Fu, J., Walker, E. Y., Yatsenko, D., Sinz, F. H., Reimer, J. & Tolias, A.S. A global map of orientation tuning in mouse visual cortex. *bioRxiv* (2019). doi: 10.1101/745323

Girshick, A.R., Landy, M.S. & Simoncelli, E.P. Cardinal rules: visual orientation perception reflects knowledge of environmental statistics. *Nat Neurosci* **14**, 926-932 (2011).

Goltstein, P.M., Reinert, S., Bonhoeffer, T. & Hübener, M. Mouse visual cortex areas represent perceptual and semantic features of learned visual categories. *Nat Neurosci* **24**, 1441-1451 (2021).

Humble, J., Hiratsuka, K., Kasai, H. & Toyozumi, T. Intrinsic spine dynamics are critical for recurrent network learning in models with and without autism spectrum disorder. *Front Comput Neurosci* **13**, 38 (2019).

Katlowitz, K. A., Picardo, M. A. & Long, M. A. Stable sequential activity underlying the maintenance of a precisely executed skilled behavior. *Neuron* **98**, 1133-1140 (2018).

Kreile, A. K., Bonhoeffer, T. & Hübener, M. Altered visual experience induces instructive changes of orientation preference in mouse visual cortex. *J. Neurosci.* **31**, 13911-13920 (2011).

Li, A. A., Wang, F., Wu, S. & Zhang, X. Emergence of probabilistic representation in the neural network of primary visual cortex. *iScience* **25**, 103975 (2022).

- Loewenstein, Y., Kuras, A. & Rumpel, S. Multiplicative dynamics underlie the emergence of the log-normal distribution of spine sizes in the neocortex in vivo. *J. Neurosci.* **31**, 9481–9488 (2011).
- Marks, T. D. & Goard, M. J. Stimulus-dependent representational drift in primary visual cortex. *Nat Commun* **12**, 5169 (2021).
- Montijn, J. S., Meijer, G. T., Lansink, C. S. & Pennartz, C. M. Population-level neural codes are robust to single-neuron variability from a multidimensional coding perspective. *Cell reports* **16**, 2486-2498 (2016).
- Reinert, S., Hübener, M., Bonhoeffer, T. & Goltstein, P. M. Mouse prefrontal cortex represents learned rules for categorization. *Nature* **593**, 411-417 (2021).
- Rose, T., Jaepel, J., Hübener, M. & Bonhoeffer, T. Cell-specific restoration of stimulus preference after monocular deprivation in the visual cortex. *Science* **352**, 1319-1322 (2016).
- Runge, K., Cardoso, C. & De Chevigny, A. Dendritic spine plasticity: function and mechanisms. *Front Syn Neurosci* **12**, 36 (2020).
- Sheintuch, L., Rubin, A., Brande-Eilat, N., Geva, N., Sadeh, N., Pinchasof, O. & Ziv, Y. Tracking the same neurons across multiple days in Ca²⁺ imaging data. *Cell Rep* **21**, 1102-1115 (2017).
- Statman, A., Kaufman, M., Minerbi, A., Ziv, N. E. & Brenner, N. Synaptic size dynamics as an effectively stochastic process. *PLoS Comput Biol* **10**, e1003846 (2014).
- Yasumatsu, N., Matsuzaki, M., Miyazaki, T., Noguchi, J. & Kasai, H. Principles of long-term dynamics of dendritic spines. *J. Neurosci.* **28**, 13592–13608 (2008).

Additional minor corrections

- In the legend of Supplementary Fig. 2e “*Same as c...*” was corrected to “*Same as d...*”.
- We added the sex of the mice to the abstract as per Nature communications guidelines, line 20-21: “*Here, using chronic two-photon calcium imaging in primary visual cortex of female mice,*”
- Other minor changes, mostly correcting comma errors and grammar, are indicated in the tracked changes version of the revised manuscript.

REVIEWERS' COMMENTS

Reviewer #1 (Remarks to the Author):

The authors have addressed my concerns. I do think that the data shown in Rebuttal figure 1 should be included in the paper (especially since it was a concern of both reviewers). It could either be included as a supplementary figure or summary statistics for different ROI correlation thresholds could be included in the main text.

Other than that minor remaining concern, I believe the manuscript is ready for publication.

Reviewer #2 (Remarks to the Author):

The authors have responded to all my comments; I have no further concerns.

Point-by-point response

We would again like to thank the reviewers for the constructive review process and are pleased to see that our additional efforts met their expectations.

Reviewers' comments are indicated in **bold**; quotes from the main manuscript in *italics*.

Reviewer #1 (Remarks to the Author):

The authors have addressed my concerns. I do think that the data shown in Rebuttal figure 1 should be included in the paper (especially since it was a concern of both reviewers). It could either be included as a supplementary figure or summary statistics for different ROI correlation thresholds could be included in the main text.

Other than that minor remaining concern, I believe the manuscript is ready for publication.

As suggested by the reviewer, we included the figure as Supplementary Fig. 2 and refer to it in the relevant positions in the main text (line 90) and the methods in lines 445 – 449:

We verified this approach by employing an additional quantitative method. To this end, we calculated pixel-wise correlations of ROIs for a subset of our data. We plotted the distribution across all comparisons as-well-as the minimum correlation for each neuron across all time points (Supplementary Fig. 2a). Excluding neurons with a minimum ROI correlation below 0.1, 0.25 or 0.5 did not substantially change the overall drift magnitude (Supplementary Fig. 2b-e).

Reviewer #2 (Remarks to the Author):

The authors have responded to all my comments; I have no further concerns.